

# Fractionation of stable carbon isotopes during microbial propionate consumption in anoxic rice paddy soils

Ralf Conrad[1], Peter Claus[1]

[1]Max Planck Institute for Terrestrial Microbiology, Karl-von-Frisch-Str. 10, 35043 Marburg, Germany

*Correspondence to:* Ralf Conrad (Conrad@mpi-marburg.mpg.de)

**Abstract.** Propionate is an important intermediate during the breakdown of organic matter in anoxic flooded paddy soils. Since there are only few experiments on carbon isotope fractionation and the magnitude of the isotopic enrichment factors ($\varepsilon$) involved, we measured propionate conversion to acetate, $CH_4$ and $CO_2$ in anoxic paddy soils. Propionate consumption was measured using samples of paddy soil from Vercelli (Italy) and the International Rice Research Institute (IRRI, the Philippines) suspended in phosphate buffer (pH 7.0), both in the absence and presence of sulfate (gypsum), and of methyl fluoride ($CH_3F$), an inhibitor of aceticlastic methanogenesis. Under methanogenic conditions, propionate was eventually degraded to $CH_4$ with acetate being a transient intermediate. Butyrate was also a minor intermediate. Methane was mainly produced by aceticlastic methanogenesis. Propionate consumption was inhibited by $CH_3F$. Whereas butyrate and $CH_4$ were $^{13}C$-depleted relative to propionate, acetate and $CO_2$ were $^{13}C$-enriched. The isotopic enrichment factors ($\varepsilon_{prop}$) of propionate consumption, determined by Mariotti plots, were in a range of -8‰ to -3.5‰. Under sulfidogenic conditions, acetate was also transiently accumulated, but $CH_4$ production was negligible. Application of $CH_3F$ hardly affected propionate degradation and acetate accumulation. The initially produced $CO_2$ was $^{13}C$-depleted, whereas the acetate was $^{13}C$-enriched. The values of $\varepsilon_{prop}$ were -3.5‰. It is concluded that degradation of organic carbon via propionate to acetate and $CO_2$ involves only little isotope fractionation. The results further indicate a major contribution of *Syntrophobacter*-type propionate fermentation under sulfidogenic conditions and *Smithella*-type propionate fermentation under methanogenic conditions. This interpretation is consistent with data of the microbial community composition published previously for the same soils.



**1 Introduction**

Propionate is a common intermediate of organic matter degradation in anoxic paddy soils. In the absence of sulfate reduction or methanogenesis propionate may accumulate to milimolar concentrations (Conrad et al., 2014; Glissmann and Conrad, 2000; Nozoe, 1997). Under methanogenic conditions propionate is degraded by fermentation. Several different biochemical pathways are conceivable for propionate fermentation (Textor et al., 1997). The major fermentation pathways are those by *Syntrophobacter* (Boone and Bryant, 1980) and *Smithella* (Liu et al., 1999) both members of Deltaproteobacteria. *Syntrophobacter* operates the methylmalonyl-CoA pathway, which results in randomization of the carbon positions of propionate (Houwen et al., 1991). This pathway can also be found in *Desulfotomaculum* sp. and *Pelotomaculum* sp. (Chen et al., 2005; DeBok et al., 2005; Imachi et al., 2002; Plugge et al., 2002), and apparently exists in many anoxic environments (Imachi et al., 2006; Krylova et al., 1997; Schink, 1985). *Smithella*, on the other hand, operates a dismutation pathway, which does not result in randomization (DeBok et al., 2001). This pathway has also been found in many anoxic environments (Gan et al., 2012; Lueders et al., 2004; Xia et al., 2019).

Propionate degradation by randomizing *Syntrophobacter* proceeds via succinate in the following way:

$$4 \text{ propionate} + 8 \text{ } H_2O \rightarrow 4 \text{ acetate} + 4 \text{ } CO_2 + 12 \text{ } H_2 \tag{1}$$

Propionate degradation by non-randomizing *Smithella* proceeds by dismutation of propionate:

$$4 \text{ propionate} \rightarrow 2 \text{ butyrate} + 2 \text{ acetate} \tag{2}$$

Butyrate is then syntrophically converted (e.g., by *Syntrophomonas* (McInerney et al., 1981)):

$$2 \text{ butyrate} + 4 \text{ } H_2O \rightarrow 4 \text{ acetate} + 4 \text{ } H_2 \tag{3}$$

The *Smithella* pathway in total:

$$4 \text{ propionate} + 4 \text{ } H_2O \rightarrow 6 \text{ acetate} + 4 \text{ } H_2 \tag{4}$$

Propionate fermentation is thermodynamically endergonic under standard conditions and therefore, requires syntrophic microbial partners that further convert the fermentation products. Under methanogenic conditions, the syntrophic partners are methanogenic archaea, which consume the products acetate and $H_2$. Under sulfidogenic conditions sulfate-reducing bacteria replace the methanogens. Propionate can also be directly oxidized to $CO_2$ by propionate-degrading sulfate reducers. The overall reaction stoichiometry is the same for *Syntrophobacter* and *Smithella*:

$$4 \text{ propionate} + 2 \text{ } H_2O \rightarrow 7 \text{ } CH_4 + 5 \text{ } CO_2, \text{ or} \tag{5}$$

$$4 \text{ propionate} + 7 \text{ sulfate} + 11 \text{ } H^+ \rightarrow 7 \text{ } HS^- + 12 \text{ } CO_2 + 12 \text{ } H_2O \tag{6}$$

Note, that the relative production of acetate and $H_2$ is different for *Syntrophobacter* and *Smithella* fermentation, being 1:3 and 3:2, respectively. Therefore, aceticlastic methanogenesis contributes relatively more than hydrogenotrophic methanogenesis, when propionate is fermented by *Smithella* rather than *Syntrophobacter*. Under methanogenic conditions, propionate degradation in anoxic paddy soils operates close to the thermodynamic limits (Krylova and Conrad, 1998; Yao and Conrad, 2001). These restrictions are more severe for *Syntrophobacter* than for *Smithella* (Dolfing, 2013).

Using paddy soil from Italy and the Philippines Liu and coworkers (Liu et al., 2018a; Liu and Conrad, 2017) have recently shown that propionate consumption under sulfidogenic conditions is mainly achieved by *Syntrophobacter* species or other Syntrophobacteraceae, which first oxidize propionate to acetate and $CO_2$, and subsequently oxidize the accumulated acetate to $CO_2$. They also showed that *Smithella* was probably involved in



methanogenic propionate degradation. The involvement of *Smithella* has also been shown for other paddy soils
and sediments (Gan et al., 2012; Lueders et al., 2004; Xia et al., 2019). Since we used in the present study the same
soils as Liu and coworkers (Liu et al., 2018a; Liu and Conrad, 2017), we assumed that propionate degradation was
achieved by the same microorganisms.
Knowledge of carbon isotope fractionation is important for the assessment of the pathways involved in
anaerobic degradation of organic matter (Conrad, 2005; Elsner et al., 2005). The $\delta^{13}$C values of organic carbon,
acetate and propionate in various soils and sediments were found to be similar (Conrad et al., 2014). The similarity
indicates that the enrichment factors ($\varepsilon$) of the processes involved in both production and consumption of
propionate are probably small. The direct determination of $\varepsilon$ values in microbial cultures of one propionate-
producing and one propionate-consuming bacterium also showed low values (Botsch and Conrad, 2011). However,
direct determination of $\varepsilon$ values in environmental samples is missing. Therefore, we decided to measure isotope
fractionation in methanogenic and sulfidogenic paddy soil amended with propionate along with the recording of
the production of acetate, $CH_4$ and $CO_2$. We also used the treatment with methyl fluoride ($CH_3F$) to inhibit the
consumption of acetate by methanogenic archaea (Janssen and Frenzel, 1997). Recently, we determined the
microbial communities in methanogenic and sulfidogenic rice field soils, which were used for assessment of $^{13}$C
isotope fractionation during acetate consumption (Conrad et al., 2021). Here we present analogous data from the
same soil suspensions prepared for the propionate degradation experiments.

## 2 Materials and Methods

*2.1 Paddy soils and incubation conditions*

The soil samples were from the research stations in Vercelli, Italy and the International Rice research Institute
(IRRI) in the Philippines. Sampling and soil characteristics were described before (Liu et al., 2018b).
The experimental setup was exactly the same as during a previous study on acetate consumption (Conrad et
al., 2021). Paddy soil was mixed with autoclaved anoxic $H_2O$ at a ratio of 1:1 and incubated under $N_2$ at 25°C for
4 weeks. In a second incubation, paddy soil was mixed with autoclaved anoxic $H_2O$ at a ratio of 1:1, was amended
with 0.07 g $CaSO_4.2H_2O$, and then incubated under $N_2$ at 25°C for 4 weeks. These two preincubated soil slurries
were sampled and stored at -20°C for later molecular analysis (see data in Conrad et al. ( 2021)). The preincubated
soil slurries were also used (in 3 replicates) for the following incubation experiments. Two different sets of
incubations were prepared. In the first set (resulting in methanogenic conditions), 5 ml soil slurry preincubated
without sulfate was incubated at 25°C with 40 ml 20 mM potassium phosphate buffer (pH 7.0) in a 150-ml bottle
under an atmosphere of $N_2$. The bottles were the amended with (i) 5 ml $H_2O$; (ii) 5 ml $H_2O$ + 4.5 ml $CH_3F$; (iii) 5
ml 50 mM sodium propionate; (iv) 5 ml 50 mM sodium acetate + 4.5 ml $CH_3F$. In the second set (resulting in
sulfidogenic conditions), 5 ml soil slurry preincubated with sulfate was incubated at 25°C with 40 ml 20 mM
potassium phosphate buffer (pH 7.0) in a 150-ml bottle under an atmosphere of $N_2$. The amendments were the
same as above, but with the addition of 200 µl of a $CaSO_4$ suspension corresponding to a concentration of 2.5 M
(giving a final concentration of 10 mM sulfate).




### 2.2 Chemical and isotopic analyses


Chemical and isotopic analyses were performed as described in detail previously (Goevert and Conrad, 2009).
Methane was analyzed by gas chromatography (GC) with flame ionization detector. Carbon dioxide was analyzed
after conversion to $CH_4$ with a Ni catalyst. Stable isotope analyses of $^{13}C/^{12}C$ in gas samples were performed using
GC-combustion isotope ratio mass spectrometry (GC-C-IRMS). Propionate, butyrate and acetate were measured
using high-performance liquid chromatography (HPLC) linked via a Finnigan LC IsoLink to an IRMS. The
isotopic values are reported in the delta notation ($\delta^{13}C$) relative to the Vienna Peedee Belemnite standard having a
$^{13}C/^{12}C$ ratio ($R_{standard}$) of 0.01118: $\delta^{13}C = 10^3 (R_{sample}/R_{standard} - 1)$. The precision of the GC-C-IRMS was ± 0.2‰,
that of the HPLC-IRMS was ± 0.3‰.

### 2.3 Calculations


Milimolar concentrations of $CH_4$ were calculated from the mixing ratios (1 ppmv = $10^{-6}$ bar) measured in the
gas phase of the incubation bottles: 1000 ppmv $CH_4$ correspond to 0.09 µmol per ml of liquid. Note, that this is
the total amount of $CH_4$ in the gas phase relative to the liquid phase.
Fractionation factors for reaction A → B are defined after Hayes (Hayes, 1993) as:
$$\alpha_{A/B} = (\delta_A + 1000)/(\delta_B + 1000) \tag{7}$$
also expressed as $\varepsilon \equiv 1000 (1 - \alpha)$ in permil. The carbon isotope enrichment factor $\varepsilon_{prop}$ associated with propionate
consumption was calculated from the temporal change of $\delta^{13}C$ of propionate as described by Mariotti et al.
(Mariotti et al., 1981) from the residual reactant
$$\delta_r = \delta_{ri} + \varepsilon [\ln(1-f)] \tag{8}$$
where $\delta_{ri}$ is the isotopic composition of the reactant (propionate) at the beginning, and $\delta_r$ is the isotopic composition
of the residual propionate, both at the instant when $f$ is determined. $f_{prop}$ is the fractional yield of the products based
on the consumption of propionate ($0 < f_{prop} < 1$). Linear regression of $\delta^{13}C$ of propionate against $\ln(1 - f)$ yields
$\varepsilon_{prop}$ as the slope of best fit lines. The regressions of $\delta^{13}C$ of propionate were done for data in the range of $f_{prop} <$
0.7. The linear regressions were done individually for each experimental replicate (n = 3) and were only accepted
if $r^2 > 0.9$. The $\varepsilon$ values resulting from the replicate experiments were then averaged (± SE).
The fraction ($f_{H2}$) of $CH_4$ derived from hydrogenotrophic methanogenesis was determined as described before
(Conrad et al., 2010) using
$$f_{H2} = (\delta^{13}C_{CH4} - \delta^{13}C_{CH4-ma})/(\delta^{13}C_{CH4-mc} - \delta^{13}C_{CH4-ma}) \tag{9}$$
with $\delta^{13}C_{CH4} = \delta^{13}C$ of total $CH_4$ produced, $\delta^{13}C_{CH4-mc} = \delta^{13}C$ of $CH_4$ produced from hydrogenotrophic
methanogenesis, which is equivalent to the $CH_4$ produced in the presence of $CH_3F$, and $\delta^{13}C_{CH4-ma} = \delta^{13}C$ of $CH_4$
produced from aceticlastic methanogenesis. The $\delta^{13}C_{CH4-ma}$ was approximated from the $\delta^{13}C$ of acetate in the
presence of $CH_3F$ assuming that the methyl group of acetate was depleted in $^{13}C$ by 8‰ (Conrad et al., 2014) and
that the enrichment factor ($\varepsilon_{CH4,ac-methyl}$) for $CH_4$ being produced from acetate-methyl was between 0 and -20‰.

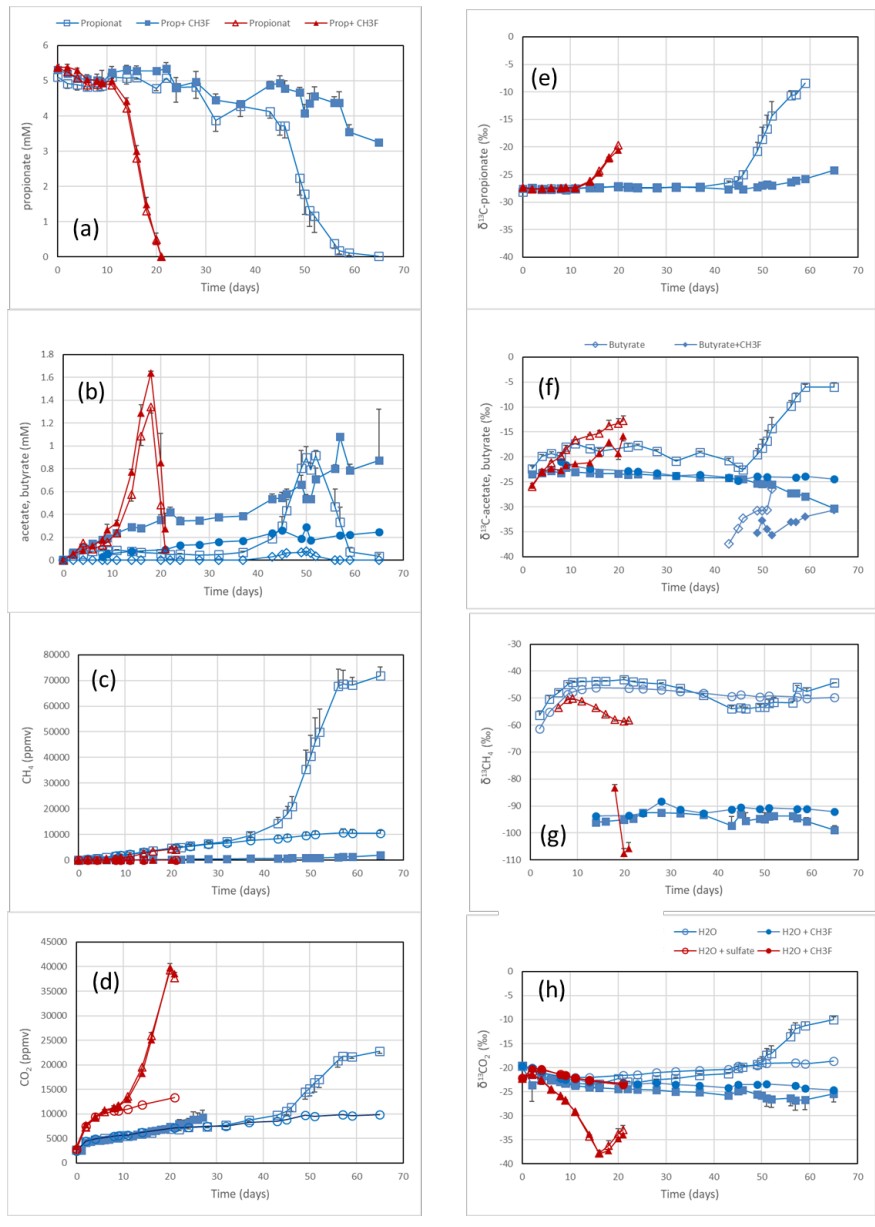

**Figure 1:** Propionate conversion to acetate, butyrate, $CH_4$ and $CO_2$ in suspensions of paddy soil from Vercelli (Italy) after addition of propionate without sulfate (blue squares) or propionate plus sulfate (gypsum) (red triangles) without $CH_3F$ (open symbols) or with $CH_3F$ (closed symbols). Controls with addition of only water (blue or red circles) are only shown occasionally. The panels show the temporal change of (a) concentrations of propionate, (b) concentrations of acetate and butyrate (blue diamonds), (c) mixing ratios of $CH_4$ (1 ppmv = $10^{-6}$ bar), (d) mixing ratios of $CO_2$, (e) $\delta^{13}C$ of propionate, (f) $\delta^{13}C$ of acetate and butyrate, (g) $\delta^{13}C$ of $CH_4$, and (h) $\delta^{13}C$ of $CO_2$. Means ± SE.





148

**3 Results**

*3.1 Conversion of propionate under methanogenic and sulfidogenic conditions*

Incubation of buffered suspensions of rice field soil from Vercelli (Fig. 1) and the IRRI (Fig. S1) resulted in
similar patterns of propionate degradation to acetate, $CH_4$ and $CO_2$. Under methanogenic conditions in the absence
of sulfate, propionate degradation started after a lag phase of about 20 d (Fig. 1a) resulting in the production of
acetate (Fig. 1b), $CH_4$ (Fig. 1c) and $CO_2$ (Fig. 1d). The formation of acetate, $CH_4$ and $CO_2$ in the absence of
propionate was only very small. The accumulation of acetate was only transient, except when aceticlastic
methanogenesis was inhibited by $CH_3F$ (Fig. 1b). Similar observations were made in IRRI soil (Fig. S1a-d). The
production of $CH_4$ was roughly equimolar to the consumption of propionate, but was nearly zero when aceticlastic
methanogenesis was inhibited by $CH_3F$ (Fig. 2a). Under these conditions, acetate accumulated to nearly equimolar
amounts with the consumed propionate (Fig. 2b), but in IRRI soil acetate accumulation was less than equimolar
(Fig. S2b). Butyrate was also a transient intermediate of propionate degradation and was produced and consumed
simultaneously with acetate (Fig. 1b, S1b). However, the accumulated concentrations were small (<0.1 mM).

In the presence of sulfate, propionate degradation started after a lag phase of only about 10 days (Fig.1a)
resulting in the accumulation of acetate (Fig. 1b) and the production of $CO_2$ (Fig. 1d), but $CH_4$ production was
close to zero (Fig. 1c). Similar results were obtained with IRRI soil (Fig. S1a-d). The accumulated acetate was
equimolar (slightly less than equimolar in the IRRI soil (Fig. S2d)) to the consumption of propionate (Fig. 2d), but
$CH_4$ was not accumulated (Fig. 2c). Addition of $CH_3F$ had no effect. Butyrate was not detected. The accumulated
acetate was subsequently degraded resulting in further production of $CO_2$ (Fig. 1b,d).

168

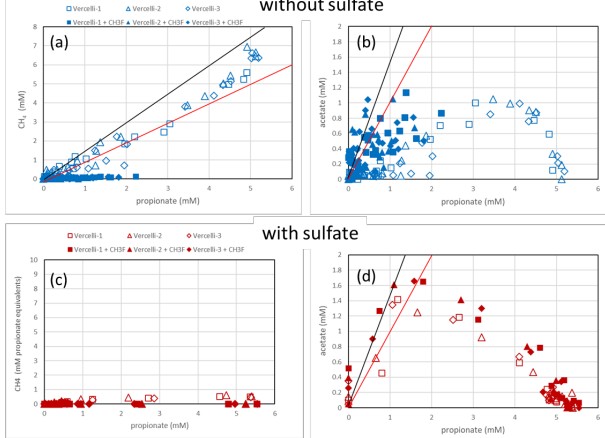



**Figure 2:** Balance of (a, c) produced $CH_4$ and (b, d) produced acetate against the consumed propionate under (a,
b) methanogenic and (c, d) sulfidogenic conditions in paddy soil from Vercelli (Italy). The open and closed
symbols denote conditions in the absence and the presence of $CH_3F$, respectively. The black and red lines in panel
(a) indicate aceticlastic methanogenesis by *Smithella* and *Syntrophobacter*, respectively. The black and red lines
in panel (b and d) indicate transient acetate production by *Smithella* and *Syntrophobacter*, respectively. The
different symbols indicate three different replicates.




*3.2 Isotope fractionation during propionate degradation*
After onset of propionate degradation, the $\delta^{13}C$ of propionate (Fig. 1e) and acetate (Fig. 1f) increased indicating
that the light isotope was preferentially consumed. The $\delta^{13}C$ values of $CO_2$ also increased (Fig. 1h). The same was
the case for butyrate (Fig. 1f). Similar results were obtained with IRRI soil (Fig. S1e-h). When aceticlastic
methanogenesis was inhibited by $CH_3F$, the $\delta^{13}C$ values of these compounds increased only slightly or decreased
(Fig. 1e,f,h). However, the $\delta^{13}C$ of $CH_4$ was much more negative (30-50‰) in the presence than in the absence of
$CH_3F$ (Fig. 1g). The $\delta^{13}C$ values of $CH_4$ in unamended soil ($H_2O$ control) were similar to those in propionate
amended soil (Fig. 1g). To visualize the change of the metabolic $^{13}C$ content of the metabolic products relative to
the substrates, the $\delta^{13}C$ values were plotted against the increasing fractions ($f_{prop}$) of propionate consumed both in
soil from Vercelli (Fig.3a) and the IRRI (Fig.3b). The patterns of $\delta^{13}C$ values against the $f_{prop}$ indicated kinetic
isotope fractionation. Note that the $\delta^{13}C$ values of acetate and $CO_2$ were higher than those of propionate, whereas
the values of butyrate and $CH_4$ were lower (Fig.3a,b). The $\delta^{13}C$ of $CH_4$ decreased until about 40% of the propionate
had been consumed, and then increased again to its initial (low) values (-50‰ to -45‰) (Fig.3a,b).
Under sulfidogenic conditions, only very little $CH_4$ was produced. Similarly as under methanogenic conditions,
the $\delta^{13}C$ of propionate (Fig. 1e) and of acetate (Fig. 1f) increased after onset of propionate degradation indicating
that the light isotope was preferentially consumed. However, the $\delta^{13}C$ values of $CO_2$ decreased during the first 10-
15 days when acetate was accumulated (Fig. 1h, S1h). Inhibition of aceticlastic methanogenesis by $CH_3F$ had no
effect on the $\delta^{13}C$ of propionate and $CO_2$, but the values of acetate increased less than in the absence of $CH_3F$ (Fig.
1f). Also, $\delta^{13}C$ of $CH_4$ was lower in the presence than in the absence of $CH_3F$ (Fig. 1g), but the amounts of $CH_4$
produced were only very small (Fig. 1c). The values of $\delta^{13}C$ of propionate and acetate increased with increasing
$f_{prop}$ (Fig. 3c,d). The $\delta^{13}C$ of acetate was generally by about 5-10‰ higher than the $\delta^{13}C$ of propionate but also
increased with $f_{prop}$ indicating kinetic isotope fractionation. However, the $\delta^{13}C$ of $CO_2$ did not increase, but instead
decreased after onset of propionate degradation reaching about -35‰ when 50% of the propionate had been
consumed and acetate accumulation had reached a maximum (Fig. 3c,d). Thereafter, $\delta^{13}C$ of $CO_2$ increased or
became constant.
Mariotti plots of the $^{13}C$ of propionate as function of $f_{prop}$ could be created for methanogenic and sulfidogenic
incubation conditions, the latter both in the absence and the presence of $CH_3F$ (Fig. 4). The lines were straight even
when more than 70% of the propionate was consumed. Nevertheless, enrichment factors ($\varepsilon$) were determined only
for $f_{prop} < 0.7$ and for regressions giving $r^2 > 0.9$. The $\varepsilon_{prop}$ values were determined for each individual incubation
and then averaged over the replicates (n = 2-3). The results for Vercelli and IRRI soils are summarized in Fig. 5.
The average $\varepsilon_{prop}$ values under methanogenic conditions were about -8‰ for Vercelli and about -3.5‰ for IRRI
soil. The average $\varepsilon_{prop}$ values under sulfidogenic conditions were around -3.5‰ in both soils and irrespectively
whether $CH_3F$ was present or not.

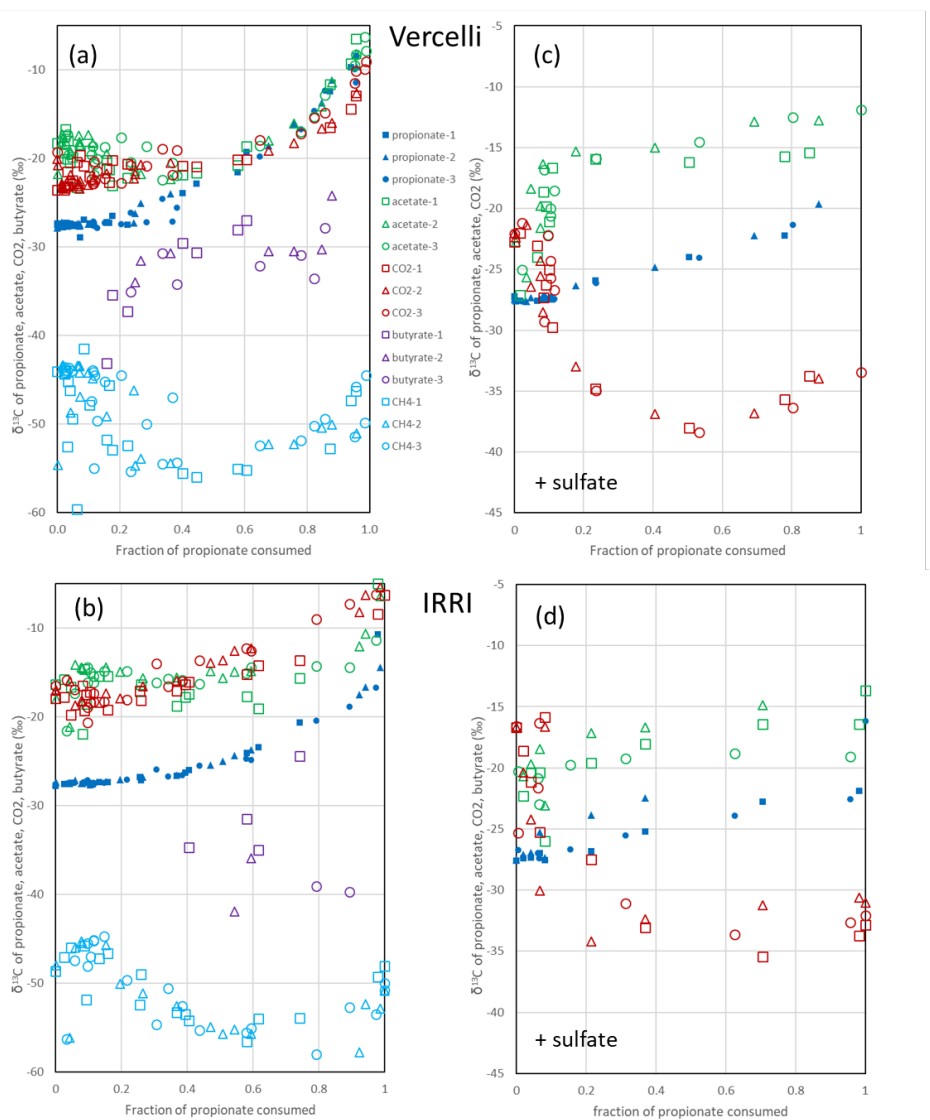

**Figure 3:** Change of $\delta^{13}C$ of propionate, acetate, butyrate, $CO_2$ and $CH_4$ relative to the fraction of propionate consumed ($f_{prop}$) under (a, b) methanogenic an (c, d) sulfidogenic conditions in paddy soil from (a, c) Vercelli (Italy) and (b, d) the IRRI (the Philippines). The different symbols indicate three different replicates.

*3.3 Hydrogenotrophic methanogenesis*

The difference in the $\delta^{13}C$ of $CH_4$ in the presence and the absence of $CH_3F$ was used together with the $\delta^{13}C$ of acetate to roughly estimate the percentage of $CH_4$ derived from $H_2/CO_2$ versus acetate (Fig. S3). The percentage fractions of hydrogenotrophic methanogenesis ($f_{H2}$) in Vercelli soil reached a maximum after 40-50 d when acetate concentrations also reached a maximum (Fig. S3a) and then decreased strongly. The same was the case in IRRI soil after around 35 d (Fig. S3b). When assuming a reasonable isotopic enrichment factor of $\varepsilon_{CH4,ac-methyl} = -15‰$,



which is in-between the $\varepsilon_{CH4,ac\text{-}methyl}$ of aceticlastic *Methanosaeta* (Penning et al., 2006; Valentine et al., 2004) and
*Methanosarcina* species (Gelwicks et al., 1994; Goevert and Conrad, 2009), the average $f_{H2}$ values were 0% for
Vercelli soil and 20% for IRRI soil (Fig. S3c).

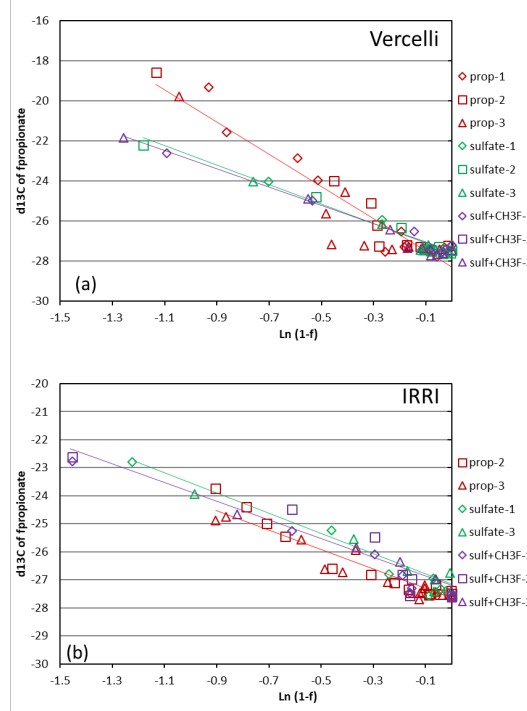



**Figure 4:** Mariotti plots of propionate consumption under methanogenic and sulfidogenic (± CH$_3$F) conditions in
paddy soil from (a) Vercelli and (b) the IRRI. The different symbols indicate three different replicates; the lines
give the results of linear regression averaged over the replicates.

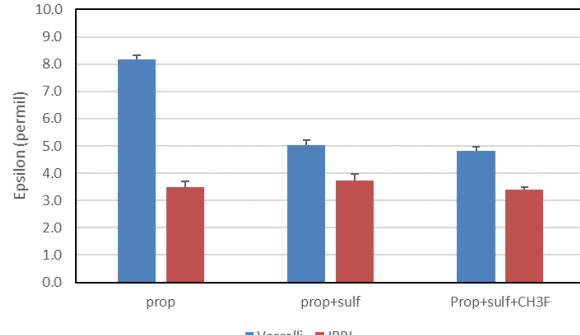


**Figure 5:** Isotopic enrichment factors ($\varepsilon_{prop}$, given as negative values) in paddy soils without and with addition of
sulfate (gypsum) and CH$_3$F. Means ± SE.




## 4 Discussion

*Pathway of propionate degradation*

Our results showed that propionate was degraded via acetate as main transient intermediate finally resulting in
the production of $CH_4$ and $CO_2$ under methanogenic and $CO_2$ under sulfidogenic conditions. These results are
consistent with previous observations by Liu and Conrad (Liu and Conrad, 2017) using the same paddy soils.
Stable isotope probing and correlation network analysis of the microbial communities have shown that propionate
degradation is accomplished by both *Syntrophopbacter* and *Smithella* species (Gan et al., 2012; Liu and Conrad,
2017; Lueders et al., 2004). The present study showed that propionate degradation under methanogenic conditions
was consistent with the major operation of the *Smithella* pathway. The main argument for this conclusion is the
observation that butyrate was a transient intermediate of propionate degradation, albeit at low concentrations (Fig.
1, S1). In the *Smithella* pathway butyrate is further fermented to acetate and $H_2$. However, production of $H_2$ is
smaller in the *Smithella* than in the *Syntrophobacter* pathway, while production of acetate is larger. Indeed,
aceticlastic methanogenesis explained all the propionate-driven methanogenesis in the paddy soils (Fig. 2a, S2a).
The average hydrogenotrophic methanogenesis by contrast contributed almost zero in Vercelli soil and only about
20% in IRRI soil (Fig. S3c). The relatively larger contribution of aceticlastic than hydrogenotrophic
methanogenesis to methanogenic propionate degradation supports the conclusion that *Smithella* pathway was
dominating over the *Syntrophobacter* pathway. Arguments against the *Smithella* pathway are that the accumulated
$CH_4$ amounted to less than the expected 1.75 mole per mole propionate consumed in Vercelli soil (Fig. 2a) and
even less in IRRI soil (Fig. S2a). With inhibition of aceticlastic methanogenesis, acetate accumulation in Vercelli
soil accounted for about 1 mole acetate per mole propionate, being in a range that is compatible with propionate
fermentation by either *Smithella* or *Syntrophobacter* (Fig. 2b). In IRRI soil however, acetate accumulation
accounted for less than 1 mole acetate per mole propionate (Fig. S2b). Note, however, that the accumulation of
acetate reflects only that part of propionate fermentation, which was not inhibited by $CH_3F$. Our conclusion that
propionate was degraded mainly by *Smithella* under methanogenic conditions is consistent with the microbial
community structure in the paddy soils from Vercelli and IRRI, which contains not only *Syntrophobacter* species
but also *Smithella* together with *Syntrophomonas*, which is able to ferment butyrate (Liu and Conrad, 2017).
Under sulfidogenic conditions, propionate can be oxidized in different ways, either directly by sulfate reducers
forming acetate and $CO_2$, or syntrophically as under methanogenic conditions, but with subsequent oxidation of
$H_2$ and acetate by sulfate reducers. Using the same paddy soils, Liu and coworkers (Liu et al., 2018a; Liu and
Conrad, 2017) recently showed that under sulfidogenic conditions propionate consumption was mainly achieved
by *Syntrophobacter* spp., which first oxidized propionate to acetate and $CO_2$, and subsequently oxidized the
accumulated acetate to $CO_2$. These were exactly the processes observed in the present study, where propionate
degradation initially resulted in almost equimolar accumulation of acetate (Fig. 2d) according to
$$4 \text{ propionate} + 3 \text{ sulfate} + 3 \text{ H}^+ \rightarrow 3 \text{ HS}^- + 4 \text{ acetate} + 4 \text{ CO}_2 + 4 \text{ H}_2\text{O} \tag{10}$$
It was interesting, that $CH_3F$ was not only a strong inhibitor of aceticlastic methanogenesis (which was
expected), but also a relatively strong inhibitor of propionate fermentation, but only under methanogenic but not
under sulfidogenic conditions. Inhibition of propionate fermentation under methanogenic conditions has been
observed before in three different paddy soils and has been interpreted as being due to the adverse thermodynamic





conditions when acetate accumulates (Conrad et al., 2014). However, this interpretation cannot be true, since
accumulation of acetate also occurred under sulfidogenic conditions, where $CH_3F$ did not inhibit propionate
degradation. In fact it is mainly the accumulation of $H_2$ rather than acetate, to which propionate degradation is
thermodynamically sensitive. This is the reason why the *Smithella* pathway is less sensitive to thermodynamic
inhibition than the *Syntrophobacter* pathway (Dolfing, 2013). However, $CH_3F$ did not inhibit $H_2$ consumption by
methanogens, as seen by the low $\delta^{13}C$ of $CH_4$ in the presence of $CH_3F$. Furthermore, the first step of the *Smithella*-
type propionate fermentation does not produce any $H_2$ and therefore, propionate should in the presence of $CH_3F$ at
least be fermented to butyrate and acetate, which however, was not the case. Hence, the reason why $CH_3F$ inhibited
propionate fermentation under methanogenic but not under sulfidogenic conditions remains unknown. Perhaps it
is *Smithella* being more sensitive to $CH_3F$ than *Syntrophobacter*.
*Fractionation during propionate degradation*
The isotopic fractionation of propionate apparently followed Raleigh distillation that is characteristic for kinetic
isotope fractionation in a closed system. The isotopic enrichment factor, which was determined from Mariotti plots,
was in the range of $\varepsilon_{prop}$ = -8‰ to -3.5‰, which is less than the enrichment factor for methanogenic acetate
consumption, which has been found to be $\varepsilon_{ac}$ = -21‰ to -17‰ (Conrad et al., 2021). The $\varepsilon_{prop}$ values are on the
same order as those predicted from $\delta^{13}C$ values of propionate, acetate and organic carbon measured in various
methanogenic soils and sediments (Conrad et al., 2014). Propionate degradation resulted in the formation of $^{13}C$-
enriched acetate and $CO_2$ and $^{13}C$-depleted butyrate and $CH_4$. The formation of $^{13}C$-depleted butyrate can be
explained by kinetic isotope effect with the preferential utilization of $^{13}C$-depleted propionate in the initial
dismutation reaction by *Smithella*. However, the production of $^{13}C$-enriched acetate cannot be explained by a linear
kinetic isotope effect. We assume that the dismutation of propionate is a branch point (Fry, 2003; Hayes, 2001), at
which the carbon flow is split into the production of $^{13}C$-enriched acetate and $^{13}C$-depleted butyrate. At the branch
point the carbon isotope flow shows a preferential flow of $^{12}C$ into the product generated by the reaction with the
larger fractionation factor, which would be butyrate. The further conversion of butyrate should produce acetate
that is depleted in $^{13}C$. This acetate together with the acetate produced from propionate dismutation should result
in the $\delta^{13}C$-acetate that is observed. The total acetate pool initially had a $\delta^{13}C$ that was up to 10‰ heavier than the
$\delta^{13}C$ of propionate. In the end, the $\delta^{13}C$ values were about equal. The observation that acetate was $^{13}C$-enriched
relative to propionate is consistent with $\delta^{13}C$ data in various soils and sediments (Conrad et al., 2014) reporting
that acetate is on the average enriched by 6‰ relative to propionate. Acetate was further converted to $CH_4$ and to
$CO_2$. In Vercelli soil, the $\delta^{13}C$ of $CH_4$ was about 25-35‰ lighter than the $\delta^{13}C$ of acetate. In IRRI soil, $^{13}C$ depletion
was even larger (30-40‰). In both soils, the isotopic enrichment factors for acetate consumption were in a range
of -12‰ to -17‰ and for $CH_4$ production from acetate in a range of -37‰ to -27‰ (Conrad et al., 2021).
Considering that a certain percentage (albeit small) of $CH_4$ was formed from $CO_2$ reduction by hydrogenotrophic
methanogenesis, which displays relatively negative enrichment factors (see the $\delta^{13}C$ of $CH_4$ in the presence of
$CH_3F$, Fig. 1g), the observed difference in $\delta^{13}C$ of $CH_4$ versus acetate is reasonable. In *Smithella* fermentation, the
only $CO_2$ production occurs during the fermentation of butyrate and the aceticlastic conversion of acetate. In both
cases $CO_2$ should be $^{13}C$-depleted relative to the substrates. Note, that this was not the case. Unfortunately, the $^{13}C$
contents of the individual C atoms of propionate, butyrate and acetate are not known. The $^{13}C$ content in the



different C positions might also affect the $\delta^{13}C$ of $CH_4$ and $CO_2$, which are formed. It is also possible that besides
*Smithella* fermentation, the *Syntrophobacter* fermentation contributed to propionate degradation. In summary, the
detailed process of isotope fractionation during the pathway of propionate degradation is unclear. However, the
magnitude of the enrichment factors involved was relatively small, being on the order of <10‰.

Under sulfidogenic conditions, propionate was most probably degraded by *Syntrophobacter* spp., first to

acetate, then finally to $CO_2$ (Liu et al., 2018a; Liu and Conrad, 2017). The carbon isotope fractionation of
propionate consumption was with an enrichment factor of $\varepsilon_{prop}$ = -3.5‰ comparatively small. Propionate was
eventually converted to two carbon products of which one was depleted (the $CO_2$) and the other was enriched (the
acetate) in $^{13}C$. In case of *Syntrophobacter*-type degradation, acetate and $CO_2$ are produced from the conversion of
pyruvate, which is generated in the methylmalonyl-CoA pathway. In this pathway, $CO_2$ is first consumed by the
conversion of propionyl-CoA to methylmalonyl-CoA and then produced by the conversion of oxaloacetate to
pyruvate. Pyruvate is finally converted to acetate and $CO_2$, which should both be $^{13}C$-depleted with respect to
pyruvate (DeNiro and Epstein, 1977). However, both acetate and $CO_2$ were initially $^{13}C$-enriched relative to
propionate (about 2-5‰), and then changed in opposite directions with acetate becoming increasingly $^{13}C$-enriched
and $CO_2$ becoming increasingly $^{13}C$-depleted until the time, when acetate accumulation had reached a maximum
(Fig. 5). Then, $\delta^{13}C$ of both acetate and $CO_2$ increased together with the increase of $^{13}C$ of propionate (Fig. 5).
Increase of $\delta^{13}C$ of acetate is often explained by consumption, especially through aceticlastic methanogenesis
(Heuer et al., 2010; Heuer et al., 2009). However, hardly any $CH_4$ was produced under sulfidogenic conditions and
the $^{13}C$ enrichment occurred during the phase of acetate accumulation. Therefore, the enrichment likely happened
during acetate production from propionate degradation. The increasing $^{13}C$-depletion of $CO_2$ can also not be
explained by consumption but only by the production from propionate. Hence, isotope fractionation during the
conversion of propionate, in particular during the conversion of propionate to pyruvate is unclear. We assume
complications during the carboxylation and decarboxylation reactions. Unfortunately, we hardly found any
literature data on the isotope fractionation of propionate fermentation. A coculture of *Syntrophobacter*
*fumaroxidans* with *Methanobacterium formicicum* exhibited marginal propionate fractionation with $\varepsilon_{prop}$ = 0.9‰
and the formation of acetate, that was slightly $^{13}C$-enriched (about 5‰) (Botsch and Conrad, 2011), similarly as
observed here. In summary, the mechanism of isotope fractionation during the conversion of propionate is not
completely clear, but the magnitude of isotope fractionation is quite low.

**5 Conclusions**
Propionate degradation under sulfidogenic conditions was explained by the metabolism of *Syntrophobacteraceae*,
which in a first step converted propionate to $^{13}C$-enriched acetate and $^{13}C$-depleted $CO_2$. By contrast, propionate
degradation under methanogenic conditions was at least partially due to metabolism by *Smithella*, which in a first
step converted propionate to $^{13}C$-enriched acetate and $^{13}C$-depleted butyrate. However, the isotopic enrichment
factors ($\varepsilon_{prop}$) of propionate consumption in two paddy soils were generally very low (-8‰ to -3.5‰) both under
methanogenic and sulfidogenic conditions. This low range is consistent with literature values of $\delta^{13}C$, collected
for propionate, acetate and organic carbon in various soils and sediments (Conrad et al., 2014). Fractionation of
propionate carbon actually seems to be smaller than fractionation of acetate, which is at least two times larger
(Conrad et al., 2021). Hence, degradation of organic carbon via propionate to acetate and $CO_2$ apparently involves



only little isotope fractionation being on the order of <10‰. By contrast, further degradation of acetate and $CO_2$
(+$H_2$) to $CH_4$ involves substantial isotope fractionation. This is also the case for chemolithotrophic acetate
production (Conrad et al., 2014).

**Supplement link**

**Author contribution:** RC designed the experiments, evaluated the data and wrote the manuscript, PC conducted
the experiments.

**Competing interests:** The authors declare that they have no conflict of interests.

**Acknowledgements**
We thank the Fonds der Chemischen Industrie for financial support.

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
