# Peer review of "Legends of supplemental figures"

_EGUsphere, 2023_

## Author Response (AR2)

Egusphere-2023-244

Reply to reviewer #1:

We thank B.Schink for his helpful comments, and reply as follows.

We intended to check the magnitude of stable isotope fractionation during anaerobic microbial propionate consumption. The overall fractionation factors were very low, and thus not very exciting. However, we think it is important to have the fractionation factors checked. In addition, patterns of isotope fractionation were different for methanogenic and sufidogenic conditions involving either Smithella or Syntrophobacter-type metabolism. We think this is an interesting observation.

l.115: change to: 'Millimolar'

l.173: We agree that the phrasing is sloppy. Both microbes produce acetate and H2 in different stoichiometries (see equ. 1-4), which are subsequently converted to CH4 by methanogens. We will replace the text: aceticlastic methanogenesis after generation of acetate by either Smithella (equ.4)  or Syntrophobacter (equ.1).

l.212: change to 'and'

l.249: change to 'that the'

l.255/56: we were also surprised by the result. However, methanogenic propionate degradation was apparently inhibited by CH3F (see Fig. 1a, e) and discussion in l. 268ff.

l.278: change to '…propionate in the presence of CH3F should…'

Reply to reviewer #2:

We thank the anonymous reviewer for helpful comments, and reply as follows:

The manuscript presents data from soil suspensions that were prepared exactly as for a previously published experiment on the isotope fractionation during acetate consumption. This is stated in the last sentences of the Introduction. Therefore, we described the methods only briefly by referring to this previous publication. However, there is no problem repeating some more details.

L.88: The main soil characteristics will be given.  The Italian soil is a sandy loam with a pH of 5.75, total C of 1.1% and total N of 0.08%. The Philippine soil is a silt loam with a pH of 6.3, total C of 1.9% and total N of 0.2%.

L.90: Anoxic water was prepared under N2

Typographical errors (mL instead of ml; CH4 with subscript) will be eliminated

L.104-112: Gas samples for analysis of partial pressures of CH4 and CO2 were taken from the headspace of the incubation bottles after vigorous manual shaking for about 30 s using a gas-tight pressure-lock syringe, which had been flushed with N2 before each sampling. Soil slurries were sampled, centrifuged

and filtered through a 0.2 µm cellulose membrane filter and stored frozen at -20C for later fatty acid analysis.

The legend of Fig. 1 did explain the symbols besides showing them in captions. This was probably overlooked by the reviewer. Therefore, no change is required.

We agree that Fig. 2 is a bit small. However, this was caused by the typesetting. We can only change the size of the axis labels if required. However, allowing more space for the typesetting would be preferable.

Figure 5: statistical analysis will be added to Fig. 5: The differences between the incubations were examined using Hukey´s post hoc test of a one-way analysis of variance (ANOVA). Different letter son top of bars indicate significant difference (P <0.05) between the data.